# Thermodynamics of Formation and Disordering of YBaCo_2_O_6-δ_ Double Perovskite as a Base for Novel Dense Ceramic Membrane Materials

**DOI:** 10.3390/membranes13010010

**Published:** 2022-12-21

**Authors:** Roman E. Yagovitin, Dmitry S. Tsvetkov, Ivan L. Ivanov, Dmitry A. Malyshkin, Vladimir V. Sereda, Andrey Yu. Zuev

**Affiliations:** Institute of Natural Sciences and Mathematics, Ural Federal University, 19 Mira St., 620002 Ekaterinburg, Russia

**Keywords:** double perovskite cobaltite, differential scanning calorimetry, oxygen nonstoichiometry, defect structure, thermochemistry, thermodynamics, solution calorimetry, enthalpy of formation

## Abstract

Differential scanning calorimetry studies of the complex oxide YBaCo_2_O_6-δ_ (YBC), combined with the literature data, allowed outlining the phase behavior of YBC depending on the oxygen content and temperature between 298 K and 773 K. The oxygen nonstoichiometry of single-phase tetragonal YBC was measured at different temperatures and oxygen partial pressures by both thermogravimetric and flow reactor methods. The defect structure of YBC was analyzed. As a result, the thermodynamic functions ($∆H_{i}^{○}$, $∆S_{i}^{○}$) of the defect reactions in YBC were determined. Experimental data on the oxygen content and those calculated based on the theoretical model were shown to be in good agreement. Standard enthalpies of formation at 298.15 K ($∆H_{f}^{○}$) were obtained for YBC depending on its oxygen content using solution calorimetry. It was found that $∆H_{f}^{○}$ = *f*(6-δ) function is linear in the range of (6-δ) from 5.018 to 5.406 and that its slope is close to the value of the enthalpy of the quasichemical reaction describing oxygen exchange between the oxide and ambient atmosphere, which confirms the reliability of the suggested defect structure model.

## 1. Introduction

Complex oxides with double perovskite structure and general formula *RE*BaCo_2_O_6-δ_, where *RE*—rare-earth metal including Y, possess high total and oxide-ion conductivity and are capable of rapid oxygen exchange, which makes these materials extremely promising for various energy conversion and storage devices—from solid oxide fuel cells (SOFCs) [1,2,3,4], including proton-conducting SOFCs with rapidly growing potential [5,6,7], to oxygen-permeating dense ceramic membranes [8,9]. While at high temperatures the membranes made of YBaCo_2_O_6-δ_ (YBC) exhibited lower oxygen flux as compared with those made of *RE*BaCo_2_O_6-δ_ with large rare-earth ions [8], this oxide has the lowest thermal expansion coefficient of all *RE*BaCo_2_O_6-δ_ (14.9·10^−6^ K^−1^ at *T* = 773–1173 K [10]). This opens the possibility of modifying YBC by doping to increase the oxygen permeability while retaining the beneficial mechanical properties. It is also worth mentioning that lowering the working temperatures of oxide membranes is a high-priority goal for enhancing their energy efficiency. It is known that in the intermediate-temperature range (673–973 K) YBC is kinetically stable and demonstrates oxygen exchange with widely variable oxygen nonstoichiometry [11,12], so it is reasonable to propose exploring the potential of using YBC as a framework for developing new lower-temperature membrane materials.

At first, however, the base YBC material should be thoroughly investigated, starting from the more fundamental and basic properties such as the phase composition and the oxygen content. The latter, on condition that the *p*O_2_-*T*-δ data set is sufficiently large and accurate, allows validating the defect structure model and evaluating the thermodynamics of defect interactions. Defect structure models are indispensable for describing and controlling many, if not most, target properties of nonstoichiometric oxides for energy applications, including the oxygen transport properties [13,14]. As with any kind of model, the best way of answering, with any certainty, the question of aptness of the defect structure model lies in finding the correlations between the modeling results and various physicochemical properties of complex oxides. For example, in our previous work such correlation was found between the model-derived partial molar enthalpy of oxygen in NdBaCo_2_O_6-δ_, $∆{\bar{h}}_{O}$, and its standard enthalpy of formation at 298.15 K vs. oxygen nonstoichiometry dependence, $∆H_{f}^{○}$(NdBaCo_2_O_6-δ_) = *f*(δ) [15]. It would be of interest to estimate similar relationships for YBC, especially taking into account that $∆H_{f}^{○}$ values themselves possess intrinsic value for fundamental thermodynamics and are necessary for further thermodynamic stability calculations.

Most of the previous studies of *RE*BaCo_2_O_6-δ_ concerned their crystal structure [16,17,18], magnetic [17,19,20] and transport [21,22] properties, while their oxygen content, defect structure and thermodynamic properties were investigated primarily at high temperatures [15,23,24,25,26], with only a few recent works focused on the intermediate-temperature range [13,15,27,28]. In this respect, the object of this work, YBC, is no exception. In the previous works, the oxygen nonstoichiometry of YBC was studied by coulometric titration at high temperatures, 1173–1323 K [29,30]. Both the *T* and *p*O_2_ ranges were necessarily restricted by the limited stability range of YBC [29,30,31]. In addition, coulometric technique employed in [30], despite being one of the high-precision nonstoichiometry measurement methods [32], can only be used at working temperatures of the solid electrolyte in the measurement cell, which are, typically, well above 900 K. As a result, the defect structure model described in [29] unavoidably contained some simplifying assumptions regarding the defect reaction thermodynamics, and the applicability of this model was limited to a very narrow oxygen content range in YBC at high temperatures.

Thereby, the present study was aimed at expanding the understanding of the defect chemistry and thermodynamics of YBC to lower temperatures. This involved obtaining the experimental *p*O_2_-*T-*δ data in the intermediate temperature range (up to 773 K) for YBC using the thermogravimetric and flow reactor methods, suitable for working at these temperatures. In order to do this with confidence that the as-measured *p*O_2_-*T-*δ data correspond to the single-phase YBC, it was necessary to assess the phase behavior of YBC between room temperature and 773 K. In addition, a correlation was to be found for YBC between the oxygen exchange thermodynamics evaluated, on the one hand, from the defect structure model, and, on the other, from the original experimental nonstoichiometry-dependent enthalpies of formation of YBC.

## 2. Experimental

The powder sample of YBC was synthesized by means of the standard ceramic technique using Y_2_O_3_, BaCO_3_ and Co_3_O_4_ as the starting materials. The purity of all the reactants was 99.99 wt %. Y_2_O_3_ had been preliminary calcined at 1373 K, and BaCO_3_ and Co_3_O_4_—at 773 K in air atmosphere to remove absorbed gases. The starting materials were mixed in an agate mortar in the amounts required by the stoichiometry of YBC, and the as-prepared mixture was annealed in air with stepwise increasing temperature from 1173 to 1373 K with intermediate regrinding. After the final annealing step at 1373 K for 24 h, the sample was rapidly cooled, to avoid the decomposition, to 773 K in air, held for 5 h in these conditions, and cooled to room temperature with a rate of 100 K·h^−1^. The phase purity of the synthesized sample was confirmed by the X-ray diffraction (XRD). The diffraction pattern was obtained by using XRD-7000 diffractometer (Shimadzu, Japan) with Cu Kα radiation.

All the thermogravimetric (TG) measurements were carried out with DynTherm LP-ST (Rubotherm, Germany) thermobalance. The absolute oxygen nonstoichiometry in the YBC samples was determined by the direct reduction in the thermobalance in H_2_/N_2_ gas mixture (1:1 by volume, 100 mL·min^−1^ total gas flow) at 1323 K. The experimental techniques are described in detail elsewhere [32]. For the sample slowly (100 K·h^−1^) cooled in air, the oxygen content, (6-δ) in YBC, was found to be 5.330 ± 0.005.

The major advantage of the flow reactor method over the other oxygen nonstoichiometry measurement techniques is its simplicity in terms of both the equipment and measurement procedure as well as the possibility of obtaining the experimental δ = *f*(*p*O_2_) dependences that closely correspond to equilibrium with the data density so high that the dependences are almost continuous. However, this is possible at sufficiently high temperatures and for the samples capable of relatively quick oxygen exchange with the atmosphere. In general, though, these conditions are not always met and, hence, continuous δ = *f*(*p*O_2_) curve cannot be obtained for all the materials and under all the conditions. Still, in such a case, stationary measurements are possible, yielding the data corresponding to the true equilibrium state of the sample but, as a drawback, providing a much lower number of points. This kind of measurement procedure was employed in this work.

The original experimental setup used for the flow reactor measurements was constructed based on the pioneering works of Starkov et al. [33], who contributed greatly to the development of the method. The schematic drawing of the setup, whose essential parts are the gas mixer, the furnace with the sample inside (or the reactor), and the temperature and pressure sensors, can also be found in Figure 1 of [33]. However, unlike in Starkov et al. [33], in the present work step-by-step relaxation procedure was realized. In this mode, a sample placed in the reactor at fixed temperature is transferred from one equilibrium state to another by switching the *p*O_2_ in the carrier gas flowing through the reactor from the *p*O_2_^(0)^ to the *p*O_2_^(1)^. At first, the reactor is heated up to the desired temperature under a constant flow (20 mL·min^−1^) of gas mixture with oxygen partial pressure *p*O_2_^(0)^. After a dwell time of around 8 h, when the oxygen content in the sample is at equilibrium, the oxygen partial pressure in the feed gas is changed from *p*O_2_^(0)^ to *p*O_2_^(1)^ by switching the ratio between the flow rates of inert gas and air. The following stepwise changes of log(*p*O_2_/atm) were implemented: −0.678 → −1.18, −0.678 → −1.54 and −0.678 → −1.59 (log is the logarithm with base 10). The gas mixtures were prepared using RRG-12 (Eltochpribor, Russia) mass-flow controllers. The total gas flow rate was 20 mL·min^−1^. The relaxation curve, i.e., *p*O_2_ at the outlet of the reactor as a function of time, is recorded at each step for further analysis using a potentiometric yttria-stabilized zirconia (YSZ) oxygen sensor. The state of equilibrium is considered to be reached when the *p*O_2_ in the outlet gas flow ceases to change and equals that in the inlet of the reactor. The experiments are performed at both several high temperatures and room temperature. The differences between the corresponding relaxation curves are due to the oxygen exchange between the sample and surrounding atmosphere. Hence, the change in the oxygen nonstoichiometry, Δδ, at each relaxation step, i.e., upon switching the *p*O_2_ from one value to another, can be calculated using the *p*O_2_ = *f*(*t*) functions obtained at room temperature, 298 K, and at a high temperature, *T*, in accordance with the equation:(1)$$∆\delta =\delta \left(1\right)-\delta \left(0\right)=\frac{2}{W}J_{\mathrm{in}}\left[\int_{0}^{t_{\mathrm{eq}}}\frac{pO_{2}\left(t\right)-pO_{2}^{\left(1\right)}}{p-pO_{2}\left(t\right)}dt-\int_{0}^{t_{\mathrm{eq}}}\frac{pO_{2}^{\prime }\left(t\right)-pO_{2}^{\left(1\right)}}{p-pO_{2}^{\prime }\left(t\right)}dt\right],$$
where δ(0) and δ(1) are the values of the oxygen nonstoichiometry in the oxide at the temperature *T* and at $pO_{2}^{\left(0\right)}$ and at $pO_{2}^{\left(1\right)}$, respectively; $pO_{2}^{\left(0\right)}$ and $pO_{2}^{\left(1\right)}$ are the starting and final oxygen partial pressures in the inlet gas mixture, respectively; $pO_{2}\left(t\right)$ and $pO_{2}^{\prime }\left(t\right)$ are the oxygen partial pressure values at the outlet of the reactor at time *t* and the temperature equal to *T* and 298 K, respectively; *t*_eq_ is the time corresponding to the end of the relaxation curve when the sample has reached the equilibrium with surrounding atmosphere; *p* is the total pressure (1 atm); *W* is the amount of the oxide in the reactor, mol; *J*_in_ is the gas flow rate, mol·s^−1^. The gas flow rate was measured with a soap film flowmeter.

The differential scanning calorimetry (DSC) studies in different atmospheres were carried out using MHTC 96 (Setaram, France) calorimeter equipped with a DSC sensor. The gas atmosphere around the sample was controlled by sweeping the calorimetric cell with mixtures of dry N_2_ (*p*O_2_ = 10^−4.3^ atm) and dry air, and *p*O_2_ at the outlet was measured by an external potentiometric YSZ oxygen sensor. The gas mixtures were prepared using RRG-12 (Eltochpribor, Russia) mass-flow controllers. The total gas flow rate was 100 mL·min^−1^. The heat sensitivity and the temperature of the DSC setup were calibrated using the heats and temperatures of fusion of high-purity standard metals (Ga, In, Sn, Pb, Al).

For the DSC experiments, two open (without lids) platinum crucibles were placed inside the DSC sensor: the sample crucible was filled with YBC powder (around 1.8 g), and the reference one—with high-purity α-alumina. The DSC measurements were performed as follows. The measurement cell was first held for 5 h at 773 K under constant flow of gas with a certain *p*O_2_, and then it was slowly (1 K·min^−1^) cooled to room temperature, followed by a two hour dwell at room temperature. Then, the DSC curve corresponding to this *p*O_2_ was obtained in the heating mode with 6 K·min^−1^ heating rate. Each DSC trace obtained as described above was reproduced 2–3 times. After that, at the final temperature of 773 K the sweeping gas mixture was changed, and the whole procedure, including the initial 5 h isothermal step, was repeated to obtain the DSC curves at a different *p*O_2_, etc. The values of the initial oxygen content in the YBC samples, i.e., of the oxygen content at room temperature after cooling in a certain atmosphere, were determined in separate TG experiments carried out with the same temperature program as in the DSC measurements.

The standard enthalpy of formation at 298.15 K of the YBC samples with different oxygen content (see Table 1) was determined using a homemade isothermal solution calorimeter. The calorimetric cell is a large-volume Dewar flask containing about 1200 mL of the solvent. The stirrer, the sample holder, the temperature sensor, which is a semiconductor thermistor, and the calorimetric heater are introduced into the calorimetric cell through a thick polyethylene foam stopper. All solution calorimetry measurements were carried out at temperature *T* = 298.15 ± 1 K. The thermometric sensitivity and the calorimetric equivalent of the calorimeter were 2.2·10^−5^ K and 0.12 J, respectively. The energy equivalent of the calorimeter was 5455 ± 10 J·K^−1^. As a solvent, 4 mol·dm^−3^ HCl solution with addition of 5 g∙l^−1^ (0.0447 mol·kg^−1^) of hydrazine dihydrochloride N_2_H_6_Cl_2_ was used. The experimental technique employed for solution calorimetry is also described elsewhere [23,24]. The reliability of the calorimeter was confirmed by measurement of the dissolution enthalpy of KCl (>99.998 wt. % purity) in water at 298.15 K. Its value, recalculated to infinite dilution, was found to be (17.26 ± 0.01) kJ·mol^−1^, which is in good agreement with the recommended value, (17.22 ± 0.04) kJ·mol^−1^ [34].

For the solution calorimetry, the powder samples of YBC with different oxygen content were prepared by quenching the YBC powders equilibrated with surrounding atmosphere under different conditions: temperatures and oxygen partial pressures. These conditions were chosen by using the *p*O_2_-*T-*δ data of YBC obtained in this and our previous works [29,30]. The values of the oxygen content in the oxide after quenching were determined by reducing the YBC samples in H_2_/N_2_ gas flow in the TG setup as described above. The as-obtained oxygen content and the corresponding annealing conditions are listed in Table 1.

In addition to YBC, the other substances and compounds used for the solution calorimetric measurements include Y metal (>99.9 wt.%, Vekton, Russia), BaCO_3_, which was pre-annealed in air 773 K for 24 h, and CoCl_2_·*n*H_2_O, obtained by recrystallization from the water solution of cobalt (II) chloride (>99 wt.%). The amount of water in CoCl_2_·*n*H_2_O was determined by decomposing it upon heating up to 1273 K in the TG setup in 50 mL·min^−1^ air flow with the heating rate of 2.2 K·min^−1^. The decomposition reaction is expressed as
(2)$$2{\mathrm{CoCl}}_{2}\cdot nH_{2}O+O_{2}=2\mathrm{CoO}+2{\mathrm{Cl}}_{2}+2nH_{2}O$$
and the amount of water *n* in CoCl_2_·*n*H_2_O was found to be equal to 4.24.

The ambient pressure during the measurements was (100 ± 4) kPa (expanded uncertainty, level of confidence 95%).

## 3. Results and Discussion

### 3.1. Sample Characterization and Phase Behavior

The XRD pattern of the YBaCo_2_O_5.33_ sample slowly (100 K·h^−1^) cooled in air is shown in Figure 1. It was indexed using the *P*4/*mmm* space group (s.g.). The refined unit cell parameters, *a* = *b* = 11.631(1) Å, *c* = 7.496(1) Å, were found to be quite consistent with those reported earlier [11].

The results of the phase behavior study of the YBC double perovskite are summarized in Figure 2 and Figure 3. As was described in Experimental, the *p*O_2_ values labelled in Figure 2 correspond simultaneously to the conditions under which the YBC samples were cooled and subsequently heated to obtain the DSC curves. Figure 2 shows that depending on the *p*O_2_, i.e., on the initial oxygen content in the YBC sample, the double perovskite exhibits from one to three phase transitions in the temperature range of 298–623 K. In addition, all YBC samples except the one with the lowest oxygen content (5.00) demonstrate oxygen release above around 623 K, which is manifested on the DSC curves as the onset of the large endothermic peak.

Because the heat flow peaks in Figure 2 tend to overlap strongly, the positions of peak maxima were chosen as the phase transition temperatures, which are plotted in Figure 3. As seen, the temperatures of both higher- and lower-temperature transitions, *T*_1_ and *T*_3_, depend on the oxygen content, whereas that of the intermediate-temperature phase transition, *T*_2_, is almost insensitive to the variation of the oxygen content in YBC.

Let us note that elucidating the exact nature of the phase transitions observed for YBC requires further thorough structural study, which is beyond the scope of this work. Nevertheless, the available literature data allow discussing the possible origins of these transitions.

Akahoshi et al. [11] reported that YBC samples with oxygen content in the range 5.25–5.44 should possess tetragonal crystal structure with the *P*4/*mmm* s.g. and 332-superstructure due to the oxygen vacancies ordering. Here, “332” means that compared to a ‘simple’ cubic perovskite unit cell, the unit cell of the double perovskite is tripled along the *a*- and *b*-axes and doubled along the *c*-axis. With increasing oxygen deficiency, the vacancy ordering is destroyed, and in the (6-δ) range of 5.15–5.19 the structure of YBC can be described by the same *P*4/*mmm* s.g. and 112-superstructure. Finally, the most oxygen-deficient YBC samples with (6-δ) close to 5.00 were found [11] to be orthorhombic with *Pmmm* space group and 112-superstructure. Similar sequence of phases depending on the oxygen content was also reported for HoBaCo_2_O_6-δ_ [35].

Figure 2 shows that the DSC results obtained in this work are in qualitative agreement with these observations [11,35]. Indeed, all the DSC traces can be divided into three groups depending on the oxygen nonstoichiometry and the supposed initial (at room temperature) crystal structure of the YBC sample. In the first group with (6-δ) from 5.28 to 5.33 all the samples show qualitatively similar patterns of three successive phase transitions, although the transition temperatures gradually change. The second and the third groups consist of one YBC sample each. For (6-δ) = 5.22, two phase transitions are distinct, and only traces of the third can be observed. In turn, for YBC with (6-δ) = 5.00 the DSC trace shows only one phase transition.

It was reported [11] that this phase transition in YBaCo_2_O_5_ (see Figure 2, (6-δ) = 5.00) is of magnetic origin and corresponds to the paramagnetic (PM) to antiferromagnetic (AFM) transition. In the samples with oxygen content higher than 5.00 the low temperature phase transition (*T*_1_ in Figure 3) seems to be also of magnetic origin. This is consistent, for example, with the change in the slope of inverse magnetic susceptibility vs. *T* curve of GdBaCo_2_O_5.38_ [36]—the double perovskite, possessing crystal structure similar to that of YBC with the same oxygen content. In this respect, it is not surprising that, as seen in Figure 2 and Figure 3, this magnetic transition is exhibited by the YBC samples in the whole wide range of oxygen content studied in this work, from (6-δ) = 5.00 to 5.33. In addition, for YBC with (6-δ) = 5.00 it was found [11] that the orthorhombic (s.g. *Pmmm*, 112-superstructure) to tetragonal (s.g. *P*4/*mmm*, 112-superstructure) phase transition occurs at the same temperature as the AFM-PM one.

The second, intermediate-temperature phase transition, which overlaps with the low-temperature one, may be identified with the transition of the tetragonal phase with s.g. *P*4/*mmm* and 332-type superstructure to the orthorhombic phase with s.g. *Pmmm* and 122-type superstructure observed in GdBaCo_2_O_5.38_ [36,37,38] at about 378–384 K due to the change in the ordering pattern of oxygen vacancies. If this interpretation is correct, then the third phase transition, undergoing at higher temperature *T*_3_, should correspond to complete disordering of oxygen vacancies and formation of the tetragonal phase with s.g. *P*4/*mmm* and 112-superstructure. This assumption is indirectly supported by the decrease in *T*_3_ with the increase in δ (see Figure 3), because it is expected that increasing the oxygen deficiency would facilitate the vacancy disordering in the oxide. Similar transition caused by the oxygen vacancy disordering was observed in GdBaCo_2_O_5.38_ [36] upon temperature increase from 611 to 787 K, although in that case it was accompanied by the oxygen release from the oxide to atmosphere.

Therefore, the DSC results obtained in the present work allow outlining the integral picture of the phase behavior of YBC depending on the oxygen content and *T*. These results will be complemented with those of neutron diffraction study, which is in progress now.

### 3.2. Oxygen Content and Thermodynamics of Disordering

As mentioned in the Introduction, the nonstoichiometry of YBC was investigated only at high temperatures 1173–1373 K in our earlier works [29,30]. Although YBC is metastable at low temperatures (e.g., below 1123 K in air [30]), we found out in this study that after at least 100 h at 773 K no detectible decomposition occurred. Between 873 K and 1073 K, however, the decomposition kinetics may not be so sluggish. For this reason, the upper temperature limit of 773 K for investigating the nonstoichiometry of YBC was chosen in the present study. In turn, at the lower-temperature limit of 573 K YBC is still capable of relatively rapid oxygen exchange and, as seen in Figure 2 and explained above, possesses tetragonal crystal structure with s.g. *P*4/*mmm* and 112-superstructure.

The experimental data on the oxygen content in single-phase tetragonal YBC measured as a function of *T* and *p*O_2_ by different techniques are shown in Figure 4. The measurements were carried at *T* = 573–773 K and *p*O_2_ = 10^−0.68^–10^−2.12^ atm, whereas the oxygen content was found to change in the wide range, from 5.398 to 5.061. As seen in Figure 4, the results obtained by TG and flow reactor methods are in good agreement with each other.

The combined *p*O_2_-*T*-δ-dataset including both the data obtained by us previously [29,30] and those measured in the present work was used to analyze the defect structure of YBC. The same defect structure model as used before [29] was employed. The defect reactions considered within the framework of this model are given in Table 2.

The expressions of equilibrium constants of these reactions along with the charge and mass balance conditions form the set of equations which was solved analytically with respect to concentrations of all the defect species involved. As a result, the model equation *p*O_2_ = *f*(*T*, δ, $∆H_{i}^{○}$, $∆S_{i}^{○}$) was obtained, where $∆H_{i}^{○}$, $∆S_{i}^{○}$ are standard enthalpies and entropies of the quasi-chemical reactions (see Table 2). This equation was fitted to the combined *p*O_2_-*T-*δ data set. The fitting results are summarized in Table 2 and Figure 5. As seen, the defect structure model describes experimental data pretty well in the whole range of temperatures and oxygen partial pressures studied.

It should also be emphasized that the values of thermodynamic parameters $∆H_{i}^{○}$, $∆S_{i}^{○}$ reassessed in this work for YBC by taking into account the lower-temperature *p*O_2_-*T-*δ data (see Table 2) became closer to those of $∆H_{i}^{○}$ and $∆S_{i}^{○}$ for GdBaCo_2_O_6-δ_ (GBC) [29]. In contrast with the double perovskite cobaltites *RE*BaCo_2_O_6-δ_ (*RE* = La, Pr, Nd, Sm, Eu) with larger rare-earth ions, both YBC and GBC are characterized by somewhat lower cobalt disproportionation enthalpy of around 20 kJ∙mol^−1^ (as compared with ≈35 kJ∙mol^−1^ for *RE*BaCo_2_O_6-δ_). In addition, YBC and GBC possess significantly lower enthalpy of oxygen vacancy localization in the rare-earth layers of the double perovskite structure, around −100 kJ∙mol^−1^ against ≈−50 kJ∙mol^−1^ for *RE*BaCo_2_O_6-δ_ [13,27]. The strongly negative values of the $V_{O}^{••}$ localization enthalpy fitted for YBC are in line with the trend revealed recently in [27] which shows increasing preference for $V_{O}^{••}$ localization in the rare-earth layers with decreasing radius of the rare-earth cation. We should emphasize here that the corresponding reaction 2 in Table 2 is not a process of the real defect cluster formation, but rather a way of accounting for the experimentally ascertained fact that in the A-site ordered layered double perovskite cobaltites the oxygen vacancies tend to locate in the rare-earth layers. When these layers are fully oxygen free the ratio between the concentrations of rare-earth ions and oxygen vacancies is exactly 1:1, hence, the concentration of the ‘quasi’ complex ${\left(Y_{Y}^{\times }V_{O}^{••}\right)}^{••}$ simply reflects the degree of deoxygenation of the rare-earth ion layers in the double perovskite structure.

### 3.3. Thermodynamics of Formation of YBaCo_2_O_6-δ_

The standard formation enthalpy of YBC at 298.15 K was calculated using the thermochemical cycle that involves the following chemical reactions:(3)$$Y_{\left(s\right)}+3{\mathrm{HCl}}_{\left(\mathrm{aq}\right)}={\mathrm{YCl}}_{3\left(\mathrm{aq}\right)}+\frac{3}{2}H_{2\left(g\right)}$$
(4)$${\mathrm{Ba}}_{\left(s\right)}+C_{\left(s\right)}+\frac{3}{2}O_{2\left(g\right)}={\mathrm{BaCO}}_{3\left(s\right)}$$
(5)$${\mathrm{BaCO}}_{3\left(s\right)}+2{\mathrm{HCl}}_{\left(\mathrm{aq}\right)}={\mathrm{BaCl}}_{2\left(\mathrm{aq}\right)}+{\mathrm{CO}}_{2\left(g\right)}+H_{2}O_{\left(\mathrm{aq}\right)}$$
(6)$${\mathrm{Co}}_{\left(s\right)}+{\mathrm{Cl}}_{2\left(g\right)}+4.24H_{2\left(g\right)}+2.12O_{2\left(g\right)}={\mathrm{CoCl}}_{2}\cdot 4.24H_{2}O_{\left(s\right)}$$
(7)$${\mathrm{CoCl}}_{2}\cdot 4.24H_{2}O_{\left(s\right)}={\mathrm{CoCl}}_{2\left(\mathrm{aq}\right)}+4.24H_{2}O_{\left(\mathrm{aq}\right)}$$
(8)$$C_{\left(s\right)}+O_{2\left(g\right)}={\mathrm{CO}}_{2\left(g\right)}$$
(9)$$H_{2\left(g\right)}+\frac{1}{2}O_{2\left(g\right)}=H_{2}O_{\left(\mathrm{aq}\right)}$$
(10)$$\frac{1}{2}H_{2\left(g\right)}+\frac{1}{2}{\mathrm{Cl}}_{2\left(g\right)}={\mathrm{HCl}}_{\left(\mathrm{aq}\right)}$$
(11)$$N_{2\left(g\right)}+3H_{2\left(g\right)}+{\mathrm{Cl}}_{2\left(g\right)}=N_{2}H_{6}{\mathrm{Cl}}_{2\left(\mathrm{aq}\right)}$$
(12)$$N_{2}H_{6}{\mathrm{Cl}}_{2\left(\mathrm{aq}\right)}+O_{2\left(g\right)}=N_{2\left(g\right)}+2H_{2}O_{\left(\mathrm{aq}\right)}+2{\mathrm{HCl}}_{\left(\mathrm{aq}\right)}$$
(13)$$Y_{\left(s\right)}+{\mathrm{Ba}}_{\left(s\right)}+2{\mathrm{Co}}_{\left(s\right)}+\left(3-\frac{\delta }{2}\right)O_{2\left(g\right)}={\mathrm{YBaCo}}_{2}O_{6-\delta \left(s\right)}$$
(14)$${\mathrm{YBaCo}}_{2}O_{6-\delta \left(s\right)}+\left(\frac{3}{4}-\frac{\delta }{2}\right)N_{2}H_{6}{\mathrm{Cl}}_{2\left(\mathrm{aq}\right)}+\left(\frac{15}{2}+\delta \right){\mathrm{HCl}}_{\left(\mathrm{aq}\right)}\;\;\;\;\;\;\;\;\;\;\;\;={\mathrm{YCl}}_{3\left(\mathrm{aq}\right)}+{\mathrm{BaCl}}_{2\left(\mathrm{aq}\right)}+2{\mathrm{CoCl}}_{2\left(\mathrm{aq}\right)}+\left(\frac{3}{4}-\frac{\delta }{2}\right)N_{2\left(g\right)}+\;\left(6-\delta \right)H_{2}O_{\left(\mathrm{aq}\right)}\;\;\;\;\;\;$$

The standard enthalpy of formation of YBC at 298.15 K corresponds to the reaction (13) and can be calculated as
(15)$$∆H_{f}^{○}=∆H_{13}^{○}=∆H_{3}+∆H_{4}^{○}+∆H_{5}+2∆H_{6}^{○}+{2∆H}_{7}-∆H_{8}^{○}-4.98∆H_{9}^{○}-4∆H_{10}^{○}+\left(\frac{3}{4}-\frac{\delta }{2}\right)∆H_{12}^{○}-∆H_{14}$$
where
(16)$$∆H_{12}^{○}=2∆H_{9}^{○}+2∆H_{10}-∆H_{11}$$

The enthalpies of reactions (3), (5), (7), (14) were determined experimentally as described in Experimental. The results of such measurements are presented in Table 3 and Table 4. The reference values of the enthalpies of formation used for the calculations are given in Table 5.

The calculated values of $∆H_{f}^{○}$(YBC) are presented in Figure 6 and Table 6. It is clearly seen that standard enthalpy of formation of YBC increases with δ. This indicates simultaneously decreasing relative thermodynamic stability of the oxide. The trend $∆H_{f}^{○}$(YBC) = *f*(6-δ) can be approximated by a straight line, as seen in Figure 6. The slope of this line, (70 ± 7) kJ·(mol O)^−1^, corresponds to the enthalpic cost of removing 1 mol of atomic oxygen from the oxide lattice and transferring it in the form of O_2_ to the surrounding atmosphere, i.e., to the enthalpy of quasichemical reaction (3) (see Table 2) describing oxygen exchange, since at room temperature the extent of cobalt disproportionation is low. This is indeed the case, as seen from Table 2 the enthalpy of quasichemical reaction (3) is equal to 63.9 ± 1.6 kJ·(mol O)^−1^. Therefore, the agreement between both values is really good.

## 4. Conclusions

The YBC double perovskite cobaltite sample, synthesized via the standard ceramic technique, was subjected to a detailed DSC study at *T* = 298–773 K in different atmospheres (*p*O_2_ = 10^−0.68^–10^−4.3^ atm). Aside from the large endothermic effects above ca. 623 K, at lower *T* the DSC curves revealed up to three phase transitions depending on the *p*O_2_, which governed the initial oxygen nonstoichiometry in YBC. The lower-temperature phase transition, exhibited by YBC with (6-δ) = 5.00–5.33 and present on all DSC traces, was identified as the one possessing magnetic origin. In turn, the higher-temperature endothermic effects on the DSC curves were tentatively ascribed to the tetragonal (*P*4/*mmm*, 332-type superstructure) to orthorhombic (*Pmmm* and 122-type superstructure) transition and a complete oxygen vacancy disordering with a new tetragonal phase formation (*P*4/*mmm*, 112-type superstructure), respectively. However, a definite assessment of the nature of all the three transitions requires a detailed neutron diffraction study.

The oxygen content in the single-phase tetragonal YBC was determined at *T* = 573–773 K and *p*O_2_ = 10^−0.68^–10^−2.12^ atm by means of the TG analysis and the flow reactor method. The new data were used for extending the range of applicability of the defect structure model [29] from the narrow high-temperature range of 1173–1323 K down to as low as 573 K. As a result, the reevaluated enthalpies and entropies of defect reactions for YBC were found to be close to those for GdBaCo_2_O_6-δ_ and distinctively different from those for *RE*BaCo_2_O_6-δ_ with larger rare-earth ions.

Standard enthalpy of formation at 298.15 K of YBC oxides with different oxygen content was calculated from the solution calorimetry results. The values of $∆H_{f}^{○}$(YBC) were shown to increase with δ, indicating the decreasing relative stability of this oxide. The slope of $∆H_{f}^{○}$(YBC)$=f\left(6-\delta \right)$ dependence is close to the value of the standard enthalpy of oxygen release that resulted from the defect structure modeling. This agreement between the model and the direct thermochemical measurement results is in favor of the validity of the defect structure model.

The results of this work will be used in the subsequent detailed structural studies, thermodynamic stability and chemical compatibility evaluations, and the investigation of defect-induced and nonstoichiometry-dependent properties of YBC and YBC-based oxides.

## Figures and Tables

**Figure 1 membranes-13-00010-f001:**
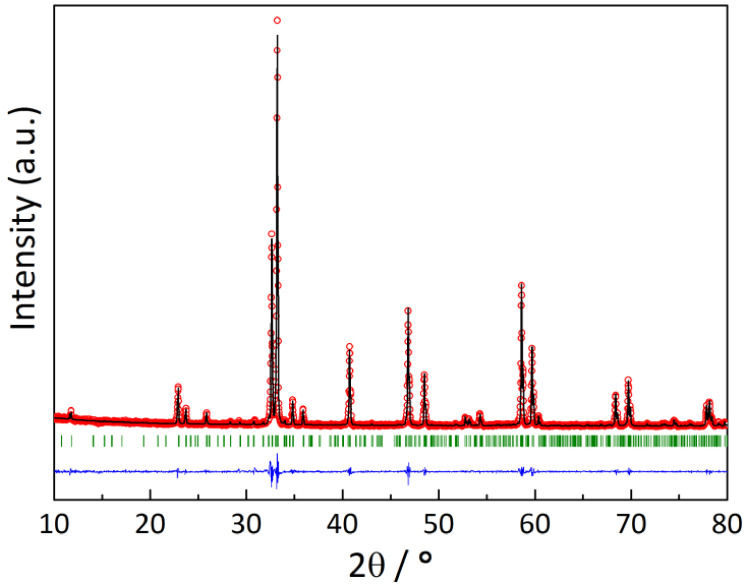
The XRD pattern of the YBaCo_2_O_5.33_ sample: red circles—experimental data, black line—calculated pattern, blue line—difference between the experimental and calculated data, green dashes—positions of the allowed Bragg reflections.

**Figure 2 membranes-13-00010-f002:**
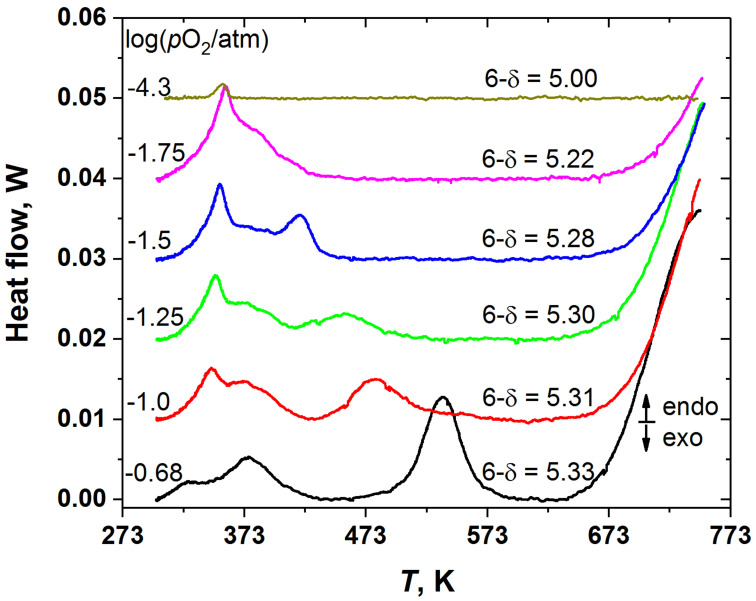
The DSC curves recorded at different *p*O_2_ for the YBC samples with different initial oxygen nonstoichiometry. The curves are shifted relative to each other for the sake of convenience.

**Figure 3 membranes-13-00010-f003:**
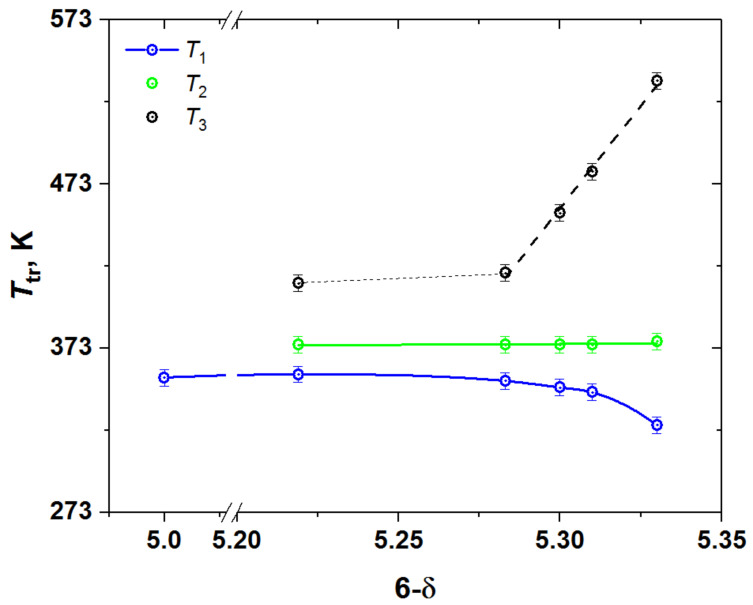
The phase transition temperatures in YBC as functions of its oxygen content.

**Figure 4 membranes-13-00010-f004:**
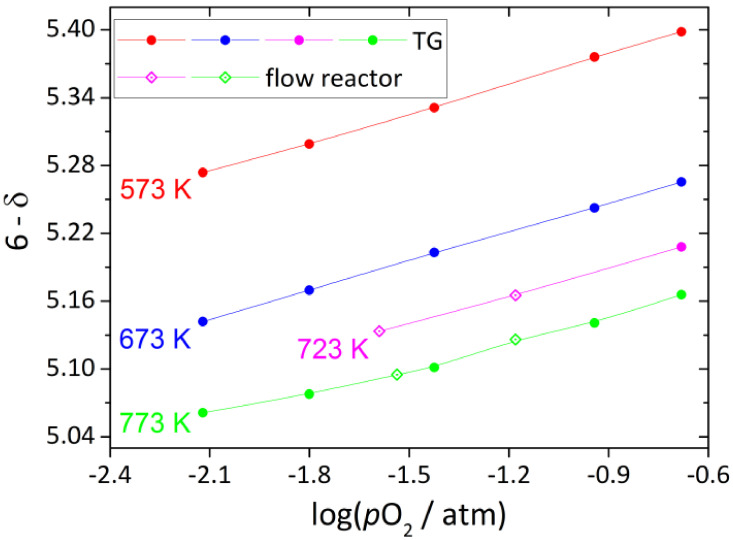
Oxygen content in YBC as function of *T* and *p*O_2_. Solid lines are guides to the eye only.

**Figure 5 membranes-13-00010-f005:**
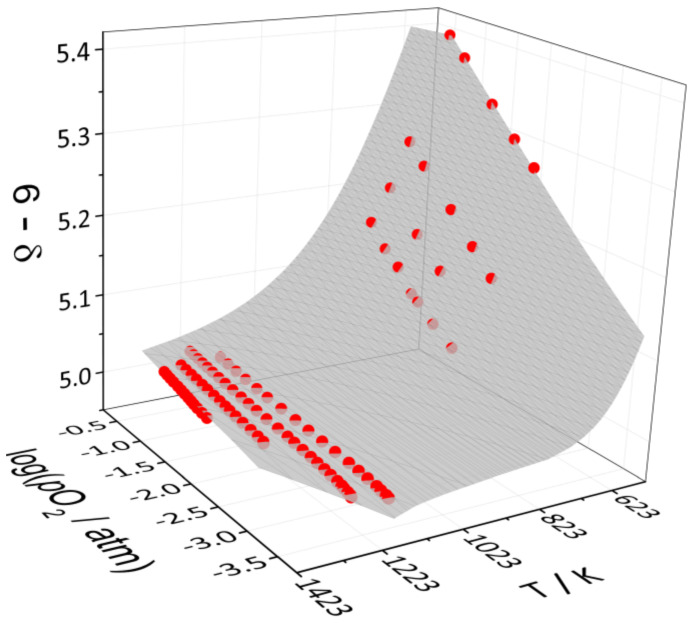
The results of fitting the defect structure model to the combined *p*O_2_-*T-*δ-dataset of YBC: the spheres correspond to the experimental values obtained in this (573–773 K) and the previous work [30] (1173–1373 K); the surface represents the model calculations.

**Figure 6 membranes-13-00010-f006:**
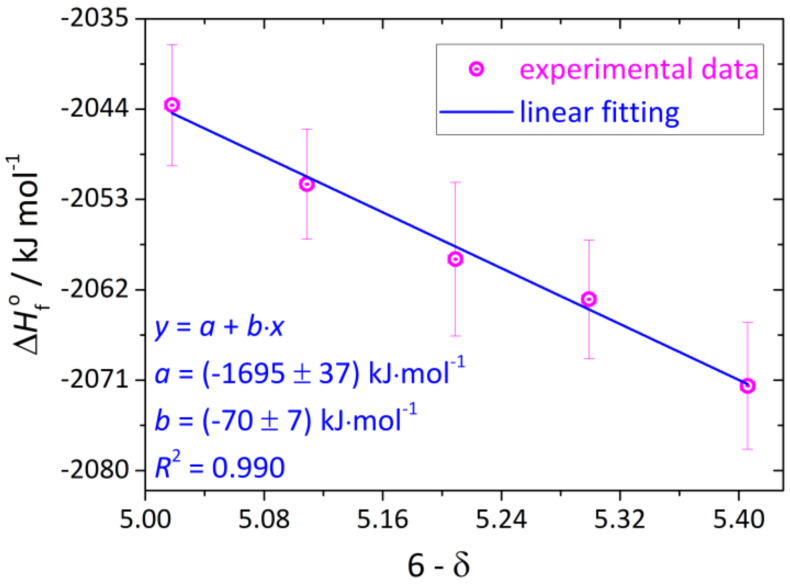
The standard formation enthalpy of YBC at 298.15 K as a function of its oxygen nonstoichiometry. Vertical and horizontal error bars correspond to expanded uncertainties at the level of confidence ≈ 95%.

**Table 1 membranes-13-00010-t001:** The annealing conditions and the oxygen content in the quenched YBC samples.

Annealing Conditions	(6-δ) in YBC
*T* = 573 K, *p*O_2_ = 0.21 atm	5.406 ± 0.005 *
*T* = 643 K, *p*O_2_ = 0.21 atm	5.299 ± 0.005
*T* = 733 K, *p*O_2_ = 0.21 atm	5.209 ± 0.005
*T* = 773 K, *p*O_2_ = 3.2 · 10^−2^ atm	5.109 ± 0.005
*T* = 1373 K, *p*O_2_ = 0.21 atm	5.018 ± 0.005

* The values after “±” sign correspond to the expanded standard uncertainties (≈95% confidence level).

**Table 2 membranes-13-00010-t002:** Results of the defect structure analysis of YBC.

Defect Reaction		$∆H_{i}^{○},$ kJ·mol^−1^	$∆S_{i}^{○},$ J·mol^−1^·K^−1^	$R^{2}$
1	$2{\mathrm{Co}}_{\mathrm{Co}}^{\times }⇆{\mathrm{Co}}_{\mathrm{Co}}^{\prime }+{\mathrm{Co}}_{\mathrm{Co}}^{•}$	18.0 ± 2.9 ^a^	0 ^b^	0.986
2	$Y_{Y}^{\times }+V_{O}^{••}⇆{\left(Y_{Y}^{\times }V_{O}^{••}\right)}^{••}$	−113.1 ± 0.9 ^a^	0 ^b^	
3	$2{\mathrm{Co}}_{\mathrm{Co}}^{\times }+O_{O}^{\times }+Y_{Y}^{\times }⇆2{\mathrm{Co}}_{\mathrm{Co}}^{\prime }+{\left(Y_{Y}^{\times }V_{O}^{••}\right)}^{••}+\frac{1}{2}O_{2}$	63.9 ± 1.6 ^a^	69.9 ± 1.0 ^a^	

^a^ The values following the «±» symbol correspond to expanded standard uncertainty calculated based on the Levenberg–Marquardt fitting procedure at the level of confidence ≈95%. ^b^ The value was fixed as 0 during the fitting procedure [15].

**Table 3 membranes-13-00010-t003:** The experimental enthalpies of solution of Y, BaCO_3_ and CoCl_2_·4.24H_2_O in 4 mol·dm^−3^ HCl with addition of N_2_H_6_Cl_2_ at 298.15 K, $∆H_{\mathrm{sol}}$.

Substance	$∆H_{sol},\;\mathrm{kJ}\cdot {\mathrm{mol}}^{-1}$	**Concentration of the Obtained Solution, mol·kg^−1^**
$Y_{\left(s\right)}$	−676 ± 3 ^a^	(2.0 ± 0.1 ^a^) · 10^−4^
${\mathrm{BaCO}}_{3\left(s\right)}$	−20.8 ± 0.4 ^a^	(1.9 ± 0.1 ^a^) · 10^−4^
${\mathrm{CoCl}}_{2}\cdot 4.24H_{2}O_{\left(s\right)}$	6.1 ± 0.4 ^a^	(3.8 ±0.2 ^a^) · 10^−4^

^a^ Values of expanded uncertainties determined at the level of confidence ≈95%.

**Table 4 membranes-13-00010-t004:** The enthalpies of solution at 298.15 K in 4 mol·dm^−3^ HCl with addition of N_2_H_6_Cl_2_, $∆H_{\mathrm{sol}}$, of YBC samples with different oxygen content.

δ in YBC	$∆H_{sol},\;\mathrm{kJ}\cdot {\mathrm{mol}}^{-1}$	Concentration of the Obtained Solution, mol·kg^−1^
0.594	−791 ± 4 ^a^	(1.8 ± 0.1 ^a^) · 10^−4^
0.701	−769 ± 3 ^a^	
0.791	−748 ± 6 ^a^	
0.891	−728 ± 4 ^a^	
0.982	−710 ± 2 ^a^	

^a^ Values of expanded uncertainties determined at the level of confidence ≈95%.

**Table 5 membranes-13-00010-t005:** Standard enthalpies of formation at 298.15 K, $∆H_{f}^{○}$, used for calculating $∆H_{f}^{○}$ (YBC).

Substance	$∆H_{f}^{○},\;\mathrm{kJ}\cdot {\mathrm{mol}}^{-1}$
BaCO_3(s)_	−1213.0 ± 0.1 ^a^ [23]
CO_2(g)_	−393.51 ± 0.05 ^a^ [39]
CoCl_2_·4.24H_2_O_(s)_	−1588.6 ± 2.1 ^a,b^ [39]
HCl_(aq)_	−162.17 ± 0.01 ^a^ [39]
H_2_O_(aq)_	−285.83 ± 0.04 ^a^ [39]
N_2_H_6_Cl_2(aq)_	−338.58 ^a^ [39]

^a^ Values of expanded uncertainties presented in papers [23,39]. ^b^ Standard enthalpy of formation of CoCl_2_·4.24H_2_O_(s)_ was calculated by using linear function $∆H_{f}^{○}\left({\mathrm{CoCl}}_{2}\cdot nH_{2}O_{\left(s\right)}\right)=f\left(n\right)$ [39].

**Table 6 membranes-13-00010-t006:** Standard formation enthalpies at 298.15 K, $∆H_{f}^{○}$, of the studied YBC samples.

(6-δ) in YBC	$∆H_{f}^{○},\;\mathrm{kJ}\cdot {\mathrm{mol}}^{-1}$
5.406	−2072 ± 7 ^a^
5.299	−2063 ± 6 ^a^
5.209	−2059 ± 8 ^a^
5.109	−2052 ± 6 ^a^
5.018	−-2044 ± 6 ^a^

^a^ Values of expanded uncertainties determined at the level of confidence ≈95%.

## Data Availability

The reported data are available by a reasonable request from the corresponding author.

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
