# Peer review of "Thermodynamics of Formation and Disordering of YBaCo2O6-δ Double Perovskite as a Base for Novel Dense Ceramic Membrane Materials"

_membranes, 2022, doi:10.3390/membranes13010010_

Round 1
Reviewer 1 Report
This is a very interesting and important paper on materials characterization in the field of solid state ionics.
It can be published after consideration of the following points:
1. In page 3/14 it is stated "After a dwell time, when the oxygen content in the sample is at equilibrium, the oxygen partial pressure in the feed gas is changed from pO2(0) to pO2(1) by switching the ratio between the flow rates of inert gas and air." The authors should give indicative dwell times so that the experiments can be repeated by interested readers.
2. In page 4/14 it is stated "he measurement cell was first held for 5 h at 773 K under constant flow of gas with a certain pO2, and then it was slowly (1 K·min-1) cooled to room temperature, followed by a several hour dwell. "
How many hours? Do the authors mean here dwell at room temperature? Please clarify!
3. Page 6/16, row 227: "....YBC requires further thorough structural study, which beyond the scope of this work. " should read YBC requires further thorough structural study, which IS beyond the scope of this work.
4. Page 9/14, row 329 "The standard formation enthalpy of YBC at 298.15 K was calculated using the thermochemical cycle that cycle involves the following chemical reactions"
delete the repeated? word "cycle" after "that"???
Author Response
Dear Editor and Reviewers,
We greatly appreciate all the reviewers’ comments and suggestions. We tried our best to address all Reviewers’ issue thoroughly. You can find our replies to Reviewers’ comments below.
Reviewer 1
This is a very interesting and important paper on materials characterization in the field of solid state ionics.
It can be published after consideration of the following points:
- In page 3/14 it is stated "After a dwell time, when the oxygen content in the sample is at equilibrium, the oxygen partial pressure in the feed gas is changed from pO2(0) to pO2(1) by switching the ratio between the flow rates of inert gas and air." The authors should give indicative dwell times so that the experiments can be repeated by interested readers.
Reply:
We agree with the Reviewer that dwell times should be given in the manuscript. The dwell times in question were around 8 hours, which is somewhat excessive but—most importantly—long enough for the oxide to reach the equilibrium. This information has been added to the manuscript.
- In page 4/14 it is stated "he measurement cell was first held for 5 h at 773 K under constant flow of gas with a certain pO2, and then it was slowly (1 K·min-1) cooled to room temperature, followed by a several hour dwell. "
How many hours? Do the authors mean here dwell at room temperature? Please clarify!
Reply:
The dwells at room temperature lasted for 2 hours. The manuscript has been corrected to clarify this.
- Page 6/16, row 227: "....YBC requires further thorough structural study, which beyond the scope of this work. " should read YBC requires further thorough structural study, which IS beyond the scope of this work.
- Page 9/14, row 329 "The standard formation enthalpy of YBC at 298.15 K was calculated using the thermochemical cycle that cycle involves the following chemical reactions"
delete the repeated? word "cycle" after "that"???
Reply:
We agree with the Reviewer that these errors should be corrected in the text of the manuscript, which has been modified accordingly. We are grateful for pointing this text imperfections.

Reviewer 2 Report
This Membrane manuscript studies the oxygen non stoichiometry in YBC double perovskite and presents an interesting set of data.
The paper would gain in clarity if corrected by a native speaker. Some sentences can be made more concise. Some articles and verbs are missing at some places.
The title does not reflect the results presented in the paper. From the title one might expect oxygen fluxes measurements.
No keywords
Page 1 line 31, Can the authors give the value of the CTE for YBC?
Experimental: Usually powders are ball milled several hours to ensure homogenous mixture of the precursors and larger batches. What size of batching and how long of a grinding in the mortar?
Page 2 lines 88-89, why is the sample rapidly cooled to 773K and held for 5 h?
Page 3: It would be useful to add a schematic of the flow reactor experiment.
Page 4 line 170: solvent 4 N HCL. What does the N stand for?
Figure 5 is difficult to read. It would be easier if the non stoichiometry was on the vertical axis, and the temperature given in K.
Tables 3 to 6, there is a dot before the word in the top left box: Substance, delta, 6-delta
Author Response
Dear Editor and Reviewers,
We greatly appreciate all the reviewers’ comments and suggestions. We tried our best to address all Reviewers’ issue thoroughly. You can find our replies to Reviewers’ comments below.
Reviewer 2
This Membrane manuscript studies the oxygen non stoichiometry in YBC double perovskite and presents an interesting set of data.
The paper would gain in clarity if corrected by a native speaker. Some sentences can be made more concise. Some articles and verbs are missing at some places.
Reply:
Indeed, none of the authors is a native speaker. Following the suggestions of the other Reviewers, we have corrected some of the linguistic errors. We also believe that the overall quality of the language in the manuscript is good enough to convey its meaning.
The title does not reflect the results presented in the paper. From the title one might expect oxygen fluxes measurements.
Reply:
We do not think that our title in its present form raises one’s expectations that the oxygen flux measurements would be performed. In our opinion, the first part of the title states that the article reports “Thermodynamics of Formation and Disordering of YBaCo2O6-δ” and the second implies just that we suggest using this material “as a base for Novel Dense Ceramic Membrane Materials”, nothing more, nothing less.
No keywords
Reply:
We agree with the Reviewer and keywords have been added to the manuscript.
Page 1 line 31, Can the authors give the value of the CTE for YBC?
Reply:
The CTE value has been added to the introduction and the manuscript has been modified accordingly.
Experimental: Usually powders are ball milled several hours to ensure homogenous mixture of the precursors and larger batches. What size of batching and how long of a grinding in the mortar?
Reply:
The batches that we used were around 10 g, so they can be easily and successfully regrinded in a mortar. Such amounts are too small for the ball mill vessels that we have. As for the time, for example, ball milling for several hours would be absolutely required to mix the precursors thoroughly if the synthesis procedure included very few or even one annealing step. In our case, there were several annealing steps with intermediate regrindings, and each regrinding in the mortar (with added ethanol to facilitate the process) took around 20 minutes. This was long enough to obtain, in the end, single-phase (i.e., by definition, homogeneous) double perovskite powder.
Page 2 lines 88-89, why is the sample rapidly cooled to 773K and held for 5 h?
Reply:
The sample was rapidly cooled to 773 K to avoid the decomposition of YBC, which might happen below 1123 K in air (please see [30] and Section 3.2), because, as we mentioned in the introduction, YBC is only kinetically stable in the intermediate-temperature range. At temperatures equal and lower 773 K the kinetic stability of YBC was found to become as high as enough for using YBC-based membrane for time which quite comparable with that of the state-of-the-art oxide ceramic membranes. Then (after reaching 773 K), the sample was held for 5 h afterwards slowly cooled to room temperature. Such treatment was employed primarily to reproduce the parameters (the dwell time and cooling) of the DSC experiments.
Page 3: It would be useful to add a schematic of the flow reactor experiment.
Reply:
The schematic drawing was present in the first drafts of the manuscript. However, after some consideration, we decided against presenting it in the final version of the manuscript. The main reason for this is that this drawing is both very basic and essentially repeats what was depicted in the article of Starkov et al. (see Ref. [33] of the manuscript submitted). We have added a sentence to Experimental to refer the interested reader to the schematic in question. The manuscript has been modified accordingly.
Page 4 line 170: solvent 4 N HCL. What does the N stand for?
Reply:
N here stands for the units of equivalent concentration, or normality. Since in this case its value is equal to the value of molarity, N has been replaced with mol·dm-3 in the manuscript.
Figure 5 is difficult to read. It would be easier if the non stoichiometry was on the vertical axis, and the temperature given in K.
Reply:
The figure has been modified as the Reviewer requested. The manuscript has been modified accordingly.
Tables 3 to 6, there is a dot before the word in the top left box: Substance, delta, 6-delta
Reply:
The dots are, indeed, unnecessary and have been removed. The manuscript has been modified accordingly.

Reviewer 3 Report
Report on « Thermodynamics of Formation and Disordering of YBaCo2O6-δ Double Perovskite”
This difficult investigation about the non-stoichiometry and the thermodynamic properties of this solid oxide is made with great care. The various calorimetric techniques and measurements are well explained and generally consistent. The data are highly interesting from a fundamental point of view. The interpretations about phase transitions are more tentative, because they will need a structural study to confirm, as correctly pointed out by the authors.
I recommend publication of this work after some questions have been addressed.
1. Page 3: “the sample was rapidly cooled to 773 K in air, held for 5 h in these conditions”. Why are 5 h chosen, is it not too short for equilibration?
2. Page 3: “For the sample slowly (100 K/h) cooled in air, the oxygen content, (6-δ) in YBC, was found to be 5.330 ± 0.005. Can the authors point out how they obtained this value, as the initial stoichiometry is very important as reference for the following study?
3. Figure 2: the third endothermal is much larger than the two others. It is attributed to a phase transition with formation of the tetragonal phase. This would imply a high phase transition enthalpy and also high entropy of transition. Is this behavior consistent with similar double perovskites, like GdBaCo2O5.38, and is there no simultaneous oxygen release observed?
4. Table 2: the very high exothermic association enthalpy of uncharged yttrium with oxide ion vacancies is surprising, because there is no electrostatic effect here (such as between positively and negatively charged vacancy pairs). Can the authors give a plausible explanation for this value (more than 1 eV !) and not only the change of cation size ?
Author Response
Dear Editor and Reviewers,
We greatly appreciate all the reviewers’ comments and suggestions. We tried our best to address all Reviewers’ issue thoroughly. You can find our replies to Reviewers’ comments below.
Reviewer 3
Report on « Thermodynamics of Formation and Disordering of YBaCo2O6-δ Double Perovskite”
This difficult investigation about the non-stoichiometry and the thermodynamic properties of this solid oxide is made with great care. The various calorimetric techniques and measurements are well explained and generally consistent. The data are highly interesting from a fundamental point of view. The interpretations about phase transitions are more tentative, because they will need a structural study to confirm, as correctly pointed out by the authors.
I recommend publication of this work after some questions have been addressed.
- Page 3: “the sample was rapidly cooled to 773 K in air, held for 5 h in these conditions”. Why are 5 h chosen, is it not too short for equilibration?
Reply:
The time of 5 h was chosen based on the thermogravimetric experiments, the results of which demonstrated that this time was long enough time for the samples to cease changing their weight and, therefore, to attain the equilibrium oxygen content.
- Page 3: “For the sample slowly (100 K/h) cooled in air, the oxygen content, (6-δ) in YBC, was found to be 5.330 ± 0.005. Can the authors point out how they obtained this value, as the initial stoichiometry is very important as reference for the following study?
Reply:
As was explained in the very same paragraph: “The absolute oxygen nonstoichiometry in the YBC samples was determined by the direct reduction in the thermobalance…”. These YBC samples included the sample slowly cooled in air, for which the oxygen content value was found to be 5.330 ± 0.005. We also presented the reference (see Ref. [32] of the manuscript submitted) where the detailed description of the reduction experiments that we performed can be found. As for the technique itself, nowadays, TG reduction is such a widespread method for the oxygen nonstoichiometry determination that its generic description can be found in other textbooks, for example, in https://doi.org/10.1016/S1573-4374(03)80008-0
- Figure 2: the third endothermal is much larger than the two others. It is attributed to a phase transition with formation of the tetragonal phase. This would imply a high phase transition enthalpy and also high entropy of transition. Is this behavior consistent with similar double perovskites, like GdBaCo2O5.38, and is there no simultaneous oxygen release observed?
Reply:
In fact, the magnitudes of all the phase transition enthalpies are rather small. This can be inferred from the y-axis, where the heat flow values are in hundredths of watt (the enthalpy is the integral of the heat flow). For the largest peak, the third peak in air atmosphere, the enthalpy of the corresponding phase transition is equal to just 0.9 ± 0.1 kJ/mol (the value after ± is twice the standard deviation of several measurement results, i.e. the expanded standard uncertainty with ~95% confidence level). The magnitude of all the other phase transition enthalpies is even lower than 0.9 kJ/mol.
For GdBaCo2O5.38, no phase transition enthalpy value was reported in the literature, so there is nothing to compare to. As to the Reviewer’s question of whether the magnitude of the phase transition enthalpy for YBC is consistent with those for similar Pmmm – P4/mmm phase transitions in the other double perovskites, our answer is: yes, it is quite consistent. For instance, as compared with around 0.9 kJ/mol for YBC, for SmBaCo2O5.52 it was 0.4–0.6 kJ/mol, for EuBaCo2O5.51 – from 1.78 to 2.18 kJ/mol, and for GdBaCo2O5.515 – 2.4 kJ/mol (please see https://doi.org/10.1021/acs.inorgchem.1c02746, https://doi.org/10.1016/j.tca.2020.178562 and the references therein). As it follows from this comparison all the values are quite small, of the order of several kJ/mol.
Regarding the last question of this comment, during the phase transition mentioned by the Reviewer no oxygen release was observed for YBC.
- Table 2: the very high exothermic association enthalpy of uncharged yttrium with oxide ion vacancies is surprising, because there is no electrostatic effect here (such as between positively and negatively charged vacancy pairs). Can the authors give a plausible explanation for this value (more than 1 eV !) and not only the change of cation size ?
Reply:
The process in question, expressed by reaction (2) from Table 2, was employed for formal describing the oxygen vacancies preferable localization in the rare-earth layers of the double perovskite structure. We have added a sentence to clarify this in the manuscript. The absence of electrostatic interactions does not play any role in this. Such preferential localization of oxygen vacancies were shown to take place in the double perovskites by means of different computational and experimental techniques (see for reference 10.1039/C3CP50316J, 10.1021/acs.inorgchem.6b02472, 10.1002/adfm.201303564, 10.1006/jssc.1998.7934, etc.). In our defect structure model, this localization is represented by the formation of Y – Vo pseudo clusters, which are not in fact the electrostatic clusters), as the ratio of rare-earth sites to oxygen sites in the rare-earth layers is exactly 1:1. Using such notation it is easy to see which vacancies are in the rare-earth layer.
As for the question of why the for REBaCo2O6-δ with the smaller rare-earth metals such as Y or Gd the vacancy localization happens to be more preferential, we think that the size factor is by itself a good explanation. Indeed, if we assume that the Vo localization in the RE layers takes place, and we know from the literature that it does, then in which oxides would it be more energetically favorable – in those with smaller RE, larger difference between Ba and RE ionic sizes and, so to speak, “more prominent” layers, or in those with larger RE, where in the limiting case of La not only the vacancies, but even the cations tend to disorder (LaBaCo2O6-δ can be obtained as a simple disordered cubic perovskite)? We think that it would be logical and quite plausible to answer “in the former” based on the ionic size considerations only.
